# Physicochemical, Nutritional and In Vitro Antidiabetic Characterisation of Blue Whiting (*Micromesistius*
*poutassou*) Protein Hydrolysates

**DOI:** 10.3390/md19070383

**Published:** 2021-07-02

**Authors:** Pádraigín A Harnedy-Rothwell, Neda Khatib, Shaun Sharkey, Ryan A Lafferty, Snehal Gite, Jason Whooley, Finbarr PM O’Harte, Richard J FitzGerald

**Affiliations:** 1Department of Biological Sciences, University of Limerick, V94 T9PX Limerick, Ireland; padraigin.harnedy@ul.ie (P.A.H.-R.); neda.khatib@ul.ie (N.K.); 2Health Research Institute, University of Limerick, V94 T9PX Limerick, Ireland; 3School of Biomedical Sciences, Ulster University, Coleraine BT52 1SA, UK; Sharkey-S5@ulster.ac.uk (S.S.); r.lafferty@ulster.ac.uk (R.A.L.); fpm.oharte@ulster.ac.uk (F.P.M.O.); 4Bio-Marine Ingredients Ireland Ltd., Lough Egish Food Park, A75 WR82 Castleblaney, Ireland; snehal@biomarine.ie (S.G.); jason@biomarine.ie (J.W.)

**Keywords:** blue whiting, protein hydrolysate, antidiabetic, type 2 diabetes mellitus, insulinotropic, functional food, amino acid analysis

## Abstract

Protein hydrolysates from low-value underutilised fish species are potential sources of high-quality dietary protein and health enhancing peptides. Six blue whiting soluble protein hydrolysates (BW-SPH-A_F), generated at industrial scale using different hydrolysis conditions, were assessed in terms of their protein equivalent content, amino acid profile and score and physicochemical properties in addition to their ability to inhibit dipeptidyl peptidase IV (DPP-IV) and stimulate the secretion of insulin from BRIN-BD11 cells. Furthermore, the effect of simulated gastrointestinal digestion (SGID) on the stability of the BW-SPHs and their associated in vitro antidiabetic activity was investigated. The BW-SPHs contained between 70–74% (*w*/*w*) protein and all essential and non-essential amino acids. All BW-SPHs mediated DPP-IV inhibitory (IC_50_: 2.12–2.90 mg protein/mL) and insulin secretory activity (2.5 mg/mL; 4.7 to 6.4-fold increase compared to the basal control (5.6 mM glucose alone)). All BW-SPHs were further hydrolysed during SGID. While the in vitro DPP-IV inhibitory and insulin secretory activity mediated by some BW-SPHs was reduced following SGID, the activity remained high. In general, the insulin secretory activity of the BW-SPHs were 4.5–5.4-fold higher than the basal control following SGID. The BW-SPHs generated herein provide potential for anti-diabetic related functional ingredients, whilst also enhancing environmental and commercial sustainability.

## 1. Introduction

Fishery resources are an excellent source of high-quality dietary protein. In general, fish meat contains between 17% and 22% (*w*/*w*) protein which is readily digestible and contains all essential amino acids in quantities that meet human dietary requirements [1,2]. Reports indicate that fish consumption per capita globally in 2016 was 19.7 kg, which equated to approximately 17.40% of the population’s intake of animal protein worldwide and approximately 6.90% of all proteins consumed [3]. However, emerging trends show an increased consumer demand in the developed world for fish-based proteins, among others, which are perceived to be healthier than terrestrial animal derived equivalents. Furthermore, with the global demand for high-quality protein rapidly increasing and the over-stretched land-based resources, emerging proteins derived from non-land based sources will be a key component and driver of food companies’ future business strategies. However, there is a growing need to increase sustainable practices within the marine sector. Fish stocks are limited and in order to assist in the protection of fisheries resources for future utilisation, annual total allowable catches (TACs) and quota allocations have been reduced over the past decade in response to growing concerns over the overutilization of numerous fish stocks [4]. Furthermore, with the landing obligation strategy of the European Common Fisheries Policy (CFP Art. 15) implemented in full across the EU in 2019, commercial fisheries have been forced to land all species caught including those that do not reach a minimum legal size or those subjected to quota. As a result, the marine industry has been forced to investigate more innovative and sustainable ways of increasing the value of all fish caught and landed [5].

Blue whiting (*Micromesistius poutassou*) is one of the most abundant fish stocks landed in the Northeast Atlantic in recent years. To date, most of the catch is reduced to fishmeal or used for fish oil as there is little or no demand for its human consumption. However, this low-value underutilized species is a rich source of high-quality protein (16.5% (*w*/*w*)) and utilization of such a resource could offer a sustainable alternative to meet the ever-increasing global protein demand, in addition to adding value to this landed resource [6]. The value can be further increased through the exploitation of blue whiting as a source of high-value health enhancing peptide ingredients for human food applications. Enzymatic hydrolysates of blue whiting proteins have shown health enhancing properties both in vitro and in vivo, in particular in the beneficial regulation of satiety and glycaemia [6,7,8,9,10]. Bioavailable peptides derived from this resource may be useful as natural alternatives, used alone or in combination with reduced doses of conventional synthetic drugs, to aid in the prevention and management of the growing prevalence of the chronic metabolic condition, Type 2 diabetes mellitus (T2DM).

One of the many challenges that exist for exploitation of fish as sources of functional ingredients is in the reproducible production of products on an industrial scale which are clean flavoured, white coloured and odour free. However, recent advances in processing technologies, in particular innovative green technologies such as ultrasound-assisted- and supercritical fluid-extraction and the availability of enzyme preparations with specific activities for enzyme assisted extraction, aid in addressing such challenges. In this study six blue whiting soluble protein hydrolysates (BW-SPHs), generated at industrial scale using different hydrolysis conditions, were assessed in terms of their protein equivalent content, amino acid profile and score, physicochemical properties, and for their ability to inhibit dipeptidyl peptidase IV (DPP-IV) and to stimulate the secretion of insulin from pancreatic BRIN-BD11 β-cells grown in culture. Furthermore, the effect of simulated gastrointestinal digestion (SGID) on the stability of the six hydrolysates and their associated in vitro antidiabetic activity was investigated.

## 2. Results

### 2.1. Protein and Amino Acid Analysis of BW-SPH-A_F

The nitrogen content of the samples ranged from 13.76% to 14.45% (*w*/*w*) and using sample specific nitrogen to protein (N:P) conversion factors (5.04 to 5.24) all six BW-SPHs (BW-SPH-A_F) generated at industrial scale were shown to have high protein equivalent contents ranging from 70.37% to 73.60% (*w*/*w*) (Table 1). The lipid content of the hydrolysates was negligible at ≤0.5% (*w*/*w*, data not shown).

In general, the total amino acid profiles of all six BW-SPHs were similar (Table 2). All six hydrolysates contained all the essential and non-essential amino acids. The amino acids found in highest abundance in all hydrolysates include Glx (combined glutamic acid and glutamine) and Asx (combined aspartic acid and asparagine) which ranged from 12.00 to 12.90 and 8.10 to 8.56 g/100 g, respectively. Amino acids found in least abundance in all hydrolysates include cysteine and tryptophan which ranged from 0.57 to 0.61 and 0.45 to 0.59 g/100 g, respectively. Hydroxyproline levels of 0.79, 0.76, 0.87, 0.61 0.79 and 0.88% (*w*/*w*), which is derived from collagen components were found in samples BW-SPH-A, BW-SPH-B, BW-SPH-C, BW-SPH-D, BW-SPH-E and BW-SPH-F, respectively (Table 2).

The free amino acid profiles of the six BW-SPHs were different (Table 2). As expected, samples BW-SPH-A, BW-SPH-B and BW-SPH-E, which had the greatest extents of hydrolysis and highest quantities of components <1 kDa, contained higher quantities of free amino acids compared to BW-SPH-C, BW-SPH-D and BW-SPH-F, which had lower degree of hydrolysis (DH) values (Table 2 and Table 3). While BW-SPH-A, BW-SPH-B and BW-SPH-E were generated using similar hydrolysis conditions, except for the second enzyme in the two-enzyme combination and had similar DH values, their free amino acid profile varied (Table 2, Table 3 and Table 4). In particular, BW-SPH-E had a lower level of free lysine at 0.31% (*w*/*w*) compared to BW-SPH-A and BW-SPH-B which had levels of 2.15 and 1.98% (*w*/*w*), respectively (Table 2). BW-SPH-A was shown to have lower levels of free isoleucine compared to the other two samples while BW-SPH-B was shown to have higher levels of free cysteine compared to BW-SPH-A and BW-SPH-E (Table 2). These differences in free amino acid profiles are most likely due to the specificity of the second enzyme preparation used for the generation of the three samples. In general, the free amino acid residues found in lower abundance in BW-SPH-C, BW-SPH-D and BW-SPH-F compared to BW-SPH-A, BW-SPH-B and BW-SPH-E include serine, Glx, valine, isoleucine, methionine and phenylalanine (Table 2). The levels of free glycine and histidine were approximately 22% and 45–65%, respectively, and was lower in BW-SPH-C compared to all other BW-SPH samples (Table 2). Furthermore, the level of free lysine in BW-SPH-C was approximately 80–90% lower than that observed in BW-SPH-A, BW-SPH-B, BW-SPH-D and BW-SPH-F.

As shown in Table 5 the calculated amino acid scores for the hydrolysates ranged from 0.36 to 0.49 when referenced against the recommended amino acid requirement pattern for infants less than 6 months, with tryptophan identified as the limiting essential amino acid in all samples [11]. For the additional two population cohorts (children from 6 months to 3 years old and children >3 years old, adolescents and adults), tryptophan was also identified as the limiting essential amino acid with amino acid scores ranging from 0.73 to 0.99 and 0.94 to 1.27 for these categories, respectively [11].

### 2.2. Physicochemical Properties of BW-SPHs Pre- and Post-Simulated Gastrointestinal Digestion (SGID)

The different hydrolysis conditions employed during industrial generation, resulted in BW-SPHs with varying DH. BW-SPH-A, BW-SPH-B, BW-SPH-E had significantly higher DH values compared to all other samples (43.0–45.8%, Table 3, *p* < 0.05) with BW-SPH-C having the lowest DH at 27.82 ± 1.11%. In general, the BW-SPHs showing high DHs contained higher quantities of components <1 kDa (Table 3). As expected, BW-SPH-C, the hydrolysate with the lowest DH had the lowest quantity of components in this molecular mass category (55.55 ± 0.13%, <1 kDa) and 2.45 ± 0.05% of components >10 kDa (Table 3). As stated previously, BW-SPH-A, BW-SPH-B and BW-SPH-E were generated using similar hydrolysis conditions apart from the second enzyme in the two-enzyme combination (Table 4). This may explain why all three samples had similar DH values and molecular mass distribution profiles. Furthermore, the lower DH value and the level of components <1 kDa observed for BW-SPH-C may be explained by the lower E:S (0.1% (*w*/*w*) minced fish) and hydrolysis time (45 min) used to generate this hydrolysate compared to that used for the generation of the other hydrolysates (E:S ≥ 0.338% (*w*/*w*) minced fish and an hydrolysis time of ≥1 h, Table 3 and Table 4).

Reversed phase-ultra-performance liquid chromatography (RP-UPLC) profiles demonstrate that the retention times of the prominent peaks eluting between 0–8 min in all six BW-SPHs were similar (Appendix A). However, the intensity of these peaks, in addition to those eluting after 10 min, varied from one sample to the other. In particular, the intensity of the peaks between 0–8 min were significantly lower in BW-SPH-C than equivalent peaks in the other samples and specifically compared to those in BW-SPH-A, BW-SPH-B and BW-SPH-F (Appendix A). In contrast the intensity of the peaks in the more hydrophobic region in particular at retention times >16 min are higher in BW-SPH-C compared to the other samples. A unique peak was also observed in BW-SPH-C at a retention time of 17.5 min. This may be associated with the lower extent of hydrolysis observed in BW-SPH-C. Furthermore, a unique peak was also seen at 15 min in the peptide profile of BW-SPH-F (Appendix A).

The DH data, molecular mass distribution and RP-UPLC profiles show that all BW-SPH were further degraded during SGID. A significant increase in DH was observed following SGID (Table 3, *p* < 0.05) with the highest increase (approximately 27.5%) observed with BW-SPH-C, the hydrolysate with the lowest DH pre-SGID. A corresponding decrease in components >1 kDa and increase in components <1 kDa, was observed with all hydrolysates (Table 3). Furthermore, on comparison of RP-UPLC peptide profiles pre- and post-SGID significant differences were observed with all BW-SPH samples (Appendix A). Similarly to that observed between BW-SPH samples pre-SGID, distinct differences were observed in the RP-UPLC profiles of samples post-SGID (BW-SPH-A to F_GI). The peak observed at a retention time of 15.0 min in the pre-SGID BW-SPH-F sample was also observed, albeit at lower intensity, in the post SGID sample (Appendix A).

### 2.3. Changes in Amino Acid Profile of the BW-SPHs as a Result of Simulated Gastrointestinal Digestion (SGID)

The free amino acid content of all BW-SPHs, with the exception of valine, increased following SGID. In particular, an increase in the level of free lysine, histidine, phenylalanine and tyrosine was observed in all samples (Table 2). Interestingly, BW-SPH-C and BW-SPH-E, which contained reduced levels of free lysine compared to the other BW-SPHs pre-SGID contained lower or similar free lysine levels to all other BW-SPHs post-SGID (Table 2). With the exception of BW-SPH-E, an increase in free arginine was observed in all BW-SPHs following SGID.

### 2.4. DPP-IV Inhibitory Activity

BW-SPH-D was shown to mediate significantly higher DPP-IV inhibitory activity (IC_50_: 2.12 ± 0.03 mg protein/mL, *p* < 0.05, Table 6) pre-SGID compared to all other BW-SPHs. BW-SPH-C showed significantly lower DPP-IV inhibitory activity (IC_50_: 2.90 ± 0.07, *p* < 0.05, Table 6) pre-SGID compared to all other BW-SPHs. A significant reduction in DPP-IV inhibitory activity was observed following SGID with BW-SPH-B, BW-SPH-D, BW-SPH-E and BW-SPH-F (*p* < 0.05, Table 6).

### 2.5. Insulin Secretory Activity

Viability analysis using the MTT assay indicated that none of the samples analysed were cytotoxic to BRIN-BD11 cells at the dose tested (2.5 mg/mL, Appendix A). All BW-SPHs samples (both pre- and post-SGID) were shown to stimulate the secretion of insulin from BRIN-BD11 cells (Table 6). Highest activity was seen with BW-SPH-F where a 6.42-fold increase in insulin secretion from BRIN-BD11 cells above the basal rate (5.6 mM glucose alone) was observed following incubation with 2.5 mg/mL of the BW-SPH. However, this activity was significant reduced following SGID with a 4.68-fold increase above the basal rate observed following incubation with an equivalent concentration of BW-SPH-F_GI (*p* < 0.05; Table 6). The insulinotropic activity of BW-SPH-D was also significantly reduced during SGID with fold changes versus the control of 6.31 ± 0.21 and 4.48 ± 0.14 recorded pre- and post-SGID, respectively. SGID had no significant effect on the insulin secretory activity of all other BW-SPHs assessed (Table 6).

## 3. Discussion

The search for new marine-derived biological sources of commercial value is a major goal for the sustainable utilisation of natural fishery resources, the utilisation of low-value by-catch and ultimately the economic survival of the fisheries industry worldwide. Valorisation of low-value fish species using a biorefinery approach, where multiple co-products arise from the biomass in a single process, represents an innovative strategy to increase the value of low-value species in addition to reducing the waste associated with processing side streams [12]. Blue whiting, a low-value underutilised fish species, which is landed in high volumes in the Northeast Atlantic, is a reservoir of high-value nutritional and health enhancing components. Blue whiting-derived proteins, in particular protein hydrolysates which have shown anti-diabetic and appetite control activities in vitro and in vivo [6,7,8,9,10], offer potential as dietary sources of protein to meet the ever-increasing global protein demand and as high-value functional peptide ingredients [6]. However, as previously stated, the reproducible production of products, which are free from marine associated tastes and odours, at large scale poses many challenges for the marine processing industry. However, recent advances in innovative technologies and processes used for the sustainable conversion of fishery resources to a range of high added-value biomaterials in a biorefinery approach has in part addressed this challenge. However, due to their predominant marine taste and odour, the oral ingestion of fish protein-derived hydrolysates generally requires the utilisation of delivery matrices containing specific masking agents to render these ingredients organoleptically acceptable for consumers. In this study six BW-SPHs generated from the same source material using different hydrolysis conditions, in a biorefinery approach at industrial scale, were investigated in terms of their protein equivalent content, amino acid profile and score, physicochemical and in vitro anti-diabetic properties and susceptibility to SGID.

To support normal bodily functions, growth and repair, humans are required to intake specific quantities of protein on a daily basis, in addition to sufficient provisions of dietary essential amino acids which are not synthesised within the body. The recommended daily protein intake for adults is 0.66 g/kg body weight with recommended levels varying depending on the protein needs of the cohort, e.g., infants, children, adolescents, pregnant and lactating women [13]. The universally used N:P conversion factor of 6.25 can only provide an estimate of a sample’s protein content. This non-specific N:P conversion factor assumes that 1 kg of protein contains 160 g of nitrogen. However, the nitrogen content of food proteins can vary significantly and is dependent on the amino acid composition, and more specifically the nitrogen content of the constituent amino acids within the food protein [14]. Therefore, calculation of sample specific N:P conversion factors is required to determine the exact protein content of a sample. N:P conversion factors for the six BW-SPHs, which were calculated based on the amino acid profiles of the test samples, ranged from 5.04 to 5.24 and were similar to those reported for other fish protein hydrolysates [15,16]. Based on these sample specific N:P conversion factors the protein equivalent content of the BW-SPHs was determined to be 70.37 to 73.60% (*w*/*w*). Furthermore, physicochemical data such as DH, molecular mass distribution, RP-UPLC and free amino profiles demonstrate that the variation in hydrolysis conditions resulted in the generation of BW-SPHs with distinctly different characteristics.

In addition to variation in the recommended daily protein intake level, different population cohorts require different quantities of each essential amino acid and as such it has been recommended that the protein quality of a food protein be determined based on three reference amino acid scoring patterns [11]. These include those set for (1) infants less than 6 months old, (2) children from 6 to 36 months old and (3) children older than 36 months, adolescents and adults [11]. In this study the ratio of each of the essential amino acids in the six BW-SPHs compared to that of levels outlined in each of the three reference cohorts were determined, with the lowest ratio being related to the amino acid score. Protein quality scores, which are evaluated based on the digestibility of the protein and the bioavailability of the essential amino acids therein, indicate whether a given food protein meets human dietary requirements for a specific cohort [11]. However, amino acid score evaluation provides useful information on how the essential amino acid levels in a protein rate against a reference protein along with information on which amino acid(s) may be limiting in the test protein sample. The six BW-SPHs studied herein were shown to contain all essential amino acids, however, when compared to the reference amino acid profiles [11] varying results were observed. In general, the level of the valine, isoleucine, leucine, tryptophan and the aromatic amino acids tyrosine and phenylalanine in the six BW-SPHs, did not meet the essential amino acid requirements for infants from birth to 6 months. The level of tryptophan in the BW-SPHs was seen to be particularly low compared to the recommended tryptophan intake levels for this cohort with amino acid scores ranging from 0.29 and 0.40. Therefore, the BW-SPHs generated herein would not be suitable for stage 1 infant formula (<6 months) applications [11]. Tryptophan was also identified as the first limiting amino acid in the 6-month to 3-year cohort, and with the exception of this amino acid, all other essential amino acids were at levels similar or higher than those of the recommended reference protein in the 6 month to 3-year cohort. It must be noted that the tryptophan level in BW-SPH-D at 8.38 mg/g protein was only slightly below the recommended 8.50 mg/g protein for this cohort [11]. However, it is highly unlikely that fish protein hydrolysates would be the sole source of protein for this grouping. Blending of BW-SPHs with other protein sources rich in tryptophan could be an option for the generation of dietary products for this cohort. Again, the first limiting amino acid in all BW-SPHs compared to the older children, adolescent, adult amino acid scoring reference was tryptophan. However, with the exception of BW-SPH-C, the amino acid scores for this cohort were all above 1, which would indicate that the levels of all essential amino acids in the other five BW-SPHs met those levels recommended for the older children, adolescent, adult grouping. The level of tryptophan in BW-SPH-C at 6.18 mg/g protein, however, meets the recommended level for adults (6.0 mg/g protein) [11].

Beyond the provision of dietary protein, protein hydrolysates, peptides and free amino acids can beneficially regulate metabolic functions in the body, including glycaemia [17,18,19]. T2DM is a chronic metabolic condition characterised by a deficiency in the secretion and/or function of insulin (i.e., insulin resistance), resulting in high blood glucose levels [20]. The prevalence of T2DM is increasing globally and effective interventions are required to prevent and manage the condition. Food protein-derived peptides with glycaemic regulatory properties may have applications in targeted dietary approaches for early-stage/prediabetic subjects where the use of drug-based treatments is not warranted. They may also have potential applications in combination with current pharmaceutical treatments thus allowing for a reduction in conventional drug dosage, which in turn could lead to reduced side-effects and healthcare costs.

The in vitro data presented herein demonstrates that the six BW-SPHs investigated can inhibit DPP-IV, the ubiquitous enzyme involved in the rapid degradation of the incretin hormones glucagon-like peptide-1 (GLP-1) and glucose-dependent insulinotropic polypeptide (GIP) in vivo and directly stimulate the secretion of insulin from pancreatic β-cells [6]. The most potent DPP-IV inhibitory activity was observed with BW-SPH-D (IC_50_: 2.12 ± 0.3 mg protein/mL). The potency values obtained for the six BW-SPHs were lower than those obtained previously for a BW-SPH generated with the proteolytic enzyme preparations Alcalase 2.4 L and Flavourzyme 500 L (IC_50_: 1.28 ± 0.4 mg /mL) [6]. However, this difference may be due to the origin of the DPP-IV enzyme utilised in both studies. The present study used human-derived DPP-IV whereas the previous study used porcine-derived DPP-IV. It has been documented that human-derived DPP-IV is less active than its porcine-derived equivalent [21,22]. For example, the IC_50_ value for the reference DPP-IV inhibitory tripeptide, Ile-Pro-Ile (Diprotin A), was 3.49 ± 0.19 and 5.00 ± 0.03 µM when using porcine and human-derived DPP-IV, respectively [21,23]. Furthermore, all hydrolysates were shown to stimulate the secretion of insulin from pancreatic BRIN-BD11 β-cells grown in culture. The highest activity was observed with BW-SPH-F, which mediated a 6.42-fold increase in insulin response, when assessed at 2.5 mg/mL, compared to the basal control (5.6 mM glucose alone). The insulinotropic results obtained herein were similar and, in some cases, higher than those obtained previously for blue whiting, boarfish and salmon protein hydrolysates when assessed at an equivalent concentration under the same conditions [6,21,24,25].

In order for a peptide to exert its activity in vivo it may need to reach the site of action in an active form. While in vivo studies can ultimately identify this, they are expensive, time-consuming, technically difficult and require ethical clearance. While in vitro digestion models have known limitations and cannot mimic the complex dynamics of the digestion process or the physiological interactions within the host in vivo they can provide information on the potential stability and metabolic fate of the active component(s) during gastrointestinal transit [26]. Physicochemical characterisation data (DH, molecular mass distribution, RP-UPLC and free amino acid profiles) showed that all six BW-SPHs were digested during SGID. Analysis of subsamples taken following exposure to pepsin indicate that little or no digestion occurred during the simulated gastric phase (data not shown). Therefore, the BW-SPH-derived peptides were mainly degraded by the intestinal enzymes trypsin, chymotrypsin and/or elastase present in Corolase PP, the enzyme preparation utilised herein to mimic intestinal digestion. SGID had no effect on the DPP-IV inhibitory activity exhibited by BW-SPH-A and BW-SPH-C. However, a significant decrease in DPP-IV inhibitory activity was observed with BW-SPH-B, BW-SPH-D, BW-SPH-E and BW-SPH-F following SGID (*p* < 0.05). This would indicate that DPP-IV inhibitory peptides in the original hydrolysate were degraded and lost activity during the SGID process. The insulinotropic potency of BW-SPH-D and BW-SPH-F was also reduced significantly (*p* < 0.05) following SGID. All other hydrolysates showed no significant reduction in insulin secretory activity following SGID. Interestingly, the DPP-IV inhibitory and insulin secretory activity mediated by BW-SPH-C_GI, the hydrolysate which was significantly degraded during SGID based on differences in DH values pre- and post-SGID, was similar to that prior to in vitro digestion. It is plausible that a combination of the following occurred: peptides mediating activity were not hydrolysed, peptides showing activity pre-SGID were degraded to fragments with less or more potent activity and fragments exhibiting activity arose from formerly inert peptides following SGID. While there was a reduction in activity following SGID with some of the BW-SPHs assessed, all samples still exhibited DPP-IV inhibitory activity following SGID (IC_50_ value range: 2.39–2.80 mg protein/mL) and exhibited high insulin secretory activity with 4.48–5.38-fold changes above the basal control observed. Therefore, these hydrolysates could potentially beneficially regulate blood glucose level in vivo through direct stimulation of insulin secretion from pancreatic β-cells or indirectly through inhibition of DPP-IV, thereby increasing or maintaining the level of circulating GLP-1 and GIP. However, this needs to be evaluated in human studies.

While the structure-activity relationship of DPP-IV inhibitory and insulin secretory peptides has not yet been fully elucidated, some structural features which influence their activity have been identified [21,27,28]. Furthermore, free amino acids such arginine, tyrosine and phenylalanine are known to inhibit DPP-IV [29,30] and glutamine, alanine, arginine, leucine, phenylalanine, valine, isoleucine and lysine are known to be strong insulin secretagogues [31,32,33,34]. However, no correlation was found between the levels of DPP-IV inhibitory and insulin stimulatory activity and the levels of amino acids both with known and unknown activity in both the total and free amino acid profiles of each of the six hydrolysates pre- and post-SGID (data not shown). Furthermore, no correlation was observed between the changes in the levels of specific free amino acids during SGID and alterations in DPP-IV inhibitory and insulin secretory activity during in vitro digestion. Crude protein hydrolysates generated with proteolytic enzymes with broad specificities can contain hundreds if not thousands of peptides having different sequences, as well as free amino acids. It is possible that the activity observed is due to the action of different peptides and amino acids working alone or more likely working synergistically. Further studies are therefore required to identify the peptides/amino acids that are potentially responsible for the observed activity through bioassay-driven fractionation of the hydrolysates and confirmatory studies need to be performed with the synthetic peptides/amino acids identified or combinations thereof.

## 4. Materials and Methods

### 4.1. Materials

Kjeldahl catalyst tablets were from VWR International (Dublin, Ireland). Low nitrogen sodium hydroxide (40% (*w*/*v*)) and sulphuric acid were from TE Laboratories Ltd. (Carlow, Ireland) and Lennox Laboratory Supplies Ltd. (Dublin, Ireland), respectively. The enzyme preparations, BC pepsin and Corolase^®^ PP, were kindly provided by Biocatalysts Ltd. (Cardiff, Wales, UK) and AB Enzymes (Darmstadt, Germany), respectively. Trinitrobenzenesulphonic acid (TNBS) reagent was from Medical Supply Co Ltd. (Dublin, Ireland). Diprotin-A (IPI: Ile-Pro-Ile) and H-Gly-Pro-7-amino-4-methyl coumarin (AMC) were purchased from Bachem Feinchemikalien (Bubendorf, Switzerland). Hank’s buffered saline solution (HBSS 10×), RPMI-1640 culture media, trypsin/EDTA (10×), penicillin-streptomycin (0.1 g/L) and fetal bovine serum (FBS) were bought from Gibco Life Technologies Ltd. (Paisley, Strathclyde, UK). Rat insulin standard and radio-labelled sodium iodide (Na ^125^I, IMS 100 mCi/mL) were obtained from Novo Industria (Copenhagen, Denmark) and Perkin Elmer (Buckinghamshire, UK), respectively. The other reagents, as well as DPP-IV (human recombinant, expressed in S*f*9 cells; ≥4500 units/μg protein), were purchased from Sigma Chemical Company Ltd. (Wicklow, Ireland).

### 4.2. Generation of Blue Whiting Soluble Protein Hydrolysates (BW-SPHs)

Six BW-SPH (BW-SPH-A_F) samples were generated at industrial scale from thawed minced blue whiting according to the parameters provided in Table 4 at BioMarine Ingredients Ireland Ltd. (Lough Egish Food Park, Castleblaney, Co., Monaghan, Ireland). All enzyme preparations used were food-grade and of microbial origin. Following hydrolysis, the enzymes were inactivated at 90 °C for 15 min and the bones were separated using a vibrating sieve. A two-phase centrifugation process was used to separate the water-soluble fraction (BW-SPH) from the undigested residue. This was followed by ultrafiltration and dehydration of the soluble phase to approx 50% (*w*/*w*) total solids, prior to spray drying.

### 4.3. Protein and Amino Acid Analysis

The moisture content of the BW-SPHs was determined gravimetrically by weight loss upon drying for 6 h at 105 °C. The nitrogen content was quantified by the procedure described by Connolly et al. [35]. Nitrogen to protein conversion factors were determined using the procedure described by Sriperm et al. [36] and employed to estimate the crude protein equivalent content. The analyses performed herein were performed in triplicate (*n* = 3). Total and free amino acid analysis was performed by an external contract supplier.

The amino acid score for each BW-SPH was calculated as: mg of limiting essential amino acid per g of test protein divided by mg of the same amino acid per g of reference protein. The reference proteins were based on the amino acid scoring patterns recommended by the FAO, which were using the reference protein for (1) infants less than 6 months old, (2) children from 6 to 36 months old and (3) children older than 36 months, adolescents and adults [11].

### 4.4. Simulated Gastrointestinal Digestion (SGID)

A modification of the method described by Walsh et al. [37] was used to perform SGID. In brief, BW-SPHs at a protein equivalent concentration of 2.0% (*w*/*v*) were incubated at 37 °C and pH 2 for 90 min with pepsin at an E:S of 2.5% (*w*/*w*). The pH was then adjusted to pH 7 with 5N NaOH and further incubated at 37 °C for 150 min with 1% (*w*/*w*) Corolase PP. Samples were then heated at 85 °C for 15 min, freeze-dried and kept at −20 °C until used.

### 4.5. Physicochemical Characterisation

The TNBS method described previously by Harnedy et al. [6] was utilsed to estimate the degree of hydrolysis (DH). The DH was calculated as follows: %DH = 100 × ((AN2 − AN1)/Npb). AN1 refers to the amino nitrogen content of the protein substrate (mg/g protein)), AN2 refers to the amino nitrogen content of the hydrolysate (mg/g protein) under investigation and Npb is the nitrogen content of the peptide bonds in the unhydrolysed protein. Values of 5.6 and 102.3 were used for AN1 and Npb, respectively [38].

Molecular mass distribution and hydrophobicity/hydrophilicity analysis were carried out by gel permeation-high performance liquid chromatography (GP-HPLC) and RP-UPLC, respectively, in order to determine the peptide profiles of the test samples [39,40]. These analyses were performed in triplicate (*n* = 3).

### 4.6. In Vitro DPP-IV Inhibitory Activity Assessment

DPP-IV inhibitory activity was ascertained as described previously [23]. In short, 50 μL of the substrate (200 μM H-Gly-Pro-AMC) was combined with 10 μL of test hydrolysate and 30 μL buffer (0.02 M Tris-HCl buffer, pH 8.0 containing 0.10 M NaCl and 1 mM EDTA) and preheated to 37 °C (5 min) in a plate reader (BioTek Synergy HT, Winooski, VT, USA). Then, 10 μL of DPP-IV was added at 8 mU/mL and the fluorescence ouput was monitored at excitation and emission wavelengths of 360 and 460 nm, respectively for 30 min. The synthetic peptide IPI (Diprotin A) was employed as a positive control at a final concnetration of 5 μM. The activity results which were performed in triplicate were expressed as IC_50_ values (inhibitory concentration (mg protein/mL) that inhibits DPP-IV by 50%).

### 4.7. Insulin Secretion Studies in Clonal Pancreatic Cells

The insulinotropic effects of samples were measured in vitro using clonal pancreatic BRIN-BD11 cells which are a hybrid cell line formed following the electrofusion of immortal RINm5F cells with a primary culture of New England Deaconess Hospital rat pancreatic islet cells [41]. The cells were cultured at 37 °C in an atmosphere of 5% CO_2_ and 95% air in RPMI-1640 tissue culture medium containing 10% (*v*/*v*) fetal calf serum, antibiotics (100 U/mL penicillin, 0.1 mg/mL streptomycin) and 11.1 mM glucose as previously described [41].

In the first instance, cellular viability of test samples (2.5 mg/mL) on BRIN-BD11 cells was determined using the MTT (3-(4,5-dimethylthiazol-2-yl)-2,5-diphenyltetrazolium bromide) assay. In brief, test samples prepared in Krebs Ringer bicarbonate buffer (KRBB) supplemented with 5.6 mM glucose were incubated for 20 min at 37 °C. KRBB was removed and Hank’s balanced salt solution (HBSS) was used to wash the cells. RPMI media (100 μL) was then added followed by 20 μL of MTT solution (5 mg/mL stock). Following a 2 h incubation in a modified atmosphere (95% O_2_, 5% CO_2_) tissue culture incubator at 37 °C the MTT/growth media was aspirated and cells were washed with HBSS. A volume of 100 μL of DMSO was used to dissolve the developed formazan crystals and the plate was agitated at room temperature for 5 min. The absorbance was read at 570 nm.

To determine the insulin secretory effects the BRIN-BD11 cells (1.5 × 10^5^ cells/well) were incubated for 20 min with the test samples in the presence of 5.6 mM glucose at 37 °C. Following incubation, 900 µL of supernatant was withdrawn and frozen at −20 °C until required. The insulin content was quantified using a dextran-coated charcoal radioimmunoassay (RIA), utilising crystalline rat insulin as a standard and guinea-pig anti-porcine antiserum (1:30,000 dilution) and ^125^I-bovine standard (10,000 cpm) [42].

### 4.8. Statistical Analysis

The software programs SPSS (Version 26, IBM Inc., Chicago, IL, USA) and GraphPad Prism version 5.0 (GraphPad Software Inc., San Diego, CA, USA) were used to perform statistical analyses on the data. Data was expressed as mean ± SD for all data except insulin secretion data which was expressed as mean ± SEM. One-way analysis of variance (ANOVA) was used to compare all values which was followed by Tukey’s and Games–Howell post hoc tests. Values before and after SGID were compared using students *t*-tests.

## 5. Conclusions

The results presented herein indicate that protein hydrolysates generated from blue whiting, a low-value underutilised fish species, provide potential as a source of high-quality protein for human nutrition. The variation in hydrolysis conditions employed led to the generation of BW-SPHs with distinctly different characteristics. The BW-SPHs were shown to mediate promising in vitro DPP-IV inhibitory and insulin secretory activity. Whilst the hydrolysates were degraded during SGID the resultant digests mediated high in vitro anti-diabetic activity which would indicate that they may have potential as anti-diabetic agents in vivo. However, the efficacy of the peptides needs to be confirmed in human studies before such claims could be made. Peptide identification and confirmatory studies are also required in order to identify the peptide(s) and amino acids responsible for the observed activity. Finally, the sustainable conversion of low-value marine resources to high-value proteinaceous ingredients such as those observed herein can help improve the fish processing industry’s financial security whilst also enhancing environmental sustainability.

## Figures and Tables

**Table 1 marinedrugs-19-00383-t001:** Total nitrogen content, nitrogen to protein (N:P) conversion factors and protein content of blue whiting (*Micromesistius poutassou*) soluble protein hydrolysates (BW-SPHs).

Sample	Nitrogen Content (g/100 g)	N:P Conversion Factor	Protein Content (g/100 g)
BW-SPH-A	14.05 ± 0.10	5.24	73.60 ± 0.53
BW-SPH-B	13.92 ± 0.05	5.19	72.24 ± 0.24
BW-SPH-C	14.45 ± 0.16	5.04	72.77 ± 0.80
BW-SPH-D	13.76 ± 0.06	5.12	70.37 ± 0.33
BW-SPH-E	13.87 ± 0.03	5.12	71.02 ± 0.14
BW-SPH-F	14.20 ± 0.06	5.15	73.03 ± 0.30

Mean ± SD (*n* = 3).

**Table 2 marinedrugs-19-00383-t002:** Total (TAA) and free amino acid (FAA) profiles of blue whiting (*Micromesistius poutassou*) soluble protein hydrolysates (BW-SPHs) pre- and post-simulated gastrointestinal digestion (GI).

	BW-SPH-A			BW-SPH-B			BW-SPH-C			BW-SPH-D			BW-SPH-E			BW-SPH-F		
	TAA	FAA	FAA-GI	TAA	FAA	FAA-GI	TAA	FAA	FAA-GI	TAA	FAA	FAA-GI	TAA	FAA	FAA-GI	TAA	FAA	FAA-GI
Asx	8.56	0.31	0.33	8.45	0.37	0.37	8.31	0.25	0.21	8.10	0.18	0.18	8.13	0.40	0.38	8.52	0.30	0.28
Ser	4.10	0.91	0.88	4.04	0.75	0.76	4.07	0.45	0.45	4.00	0.40	0.42	3.93	0.82	0.85	4.07	0.53	0.55
Glx	12.50	1.02	1.11	12.60	0.90	1.03	12.80	0.65	0.68	12.00	0.52	0.62	12.40	0.84	0.98	12.90	0.65	0.70
Gly	5.52	0.22	0.26	5.43	0.23	0.28	5.85	0.16	0.18	5.10	0.21	0.25	5.55	0.23	0.30	5.85	0.21	0.24
Arg	5.94	2.48	3.92	6.15	2.45	3.99	6.31	2.08	3.86	6.15	1.94	3.48	6.31	2.54	2.77	6.34	1.97	3.78
Pro	3.41	<0.050	<0.005	3.30	<0.050	<0.005	3.31	<0.050	<0.005	3.14	<0.050	<0.005	3.33	<0.050	<0.005	3.49	<0.050	<0.005
Ala	5.41	0.82	0.86	5.54	0.79	0.81	5.71	0.63	0.62	5.28	0.49	0.56	5.30	0.81	0.88	5.53	0.68	0.68
Cys	0.61	nd	nd	0.61	0.63	nd	0.59	0.41	nd	0.60	0.58	nd	0.57	0.23	nd	0.57	nd	nd
Tyr	3.23	0.82	2.06	2.37	0.66	1.42	2.04	0.44	1.68	2.47	0.46	1.70	2.25	0.81	1.75	2.31	0.60	1.61
Hyp	0.79	-	-	0.76	-	-	0.87	-	-	0.61	-	-	0.79	-	-	0.88	-	-
Val	3.89	0.97	0.81	3.81	1.12	0.99	3.39	0.42	0.41	3.50	0.43	0.34	3.70	1.22	1.09	3.57	0.65	0.60
Ile	3.35	0.85	0.95	3.34	1.17	1.23	2.78	0.39	0.45	2.96	0.48	0.39	3.32	1.25	1.33	3.05	0.66	0.85
Leu	5.78	2.71	2.88	5.63	3.08	3.08	5.49	1.28	2.00	5.51	1.62	1.88	5.40	3.41	3.50	5.61	2.21	2.49
Thr	3.70	0.91	0.98	3.57	0.89	1.01	3.41	0.72	0.72	3.38	0.57	0.59	3.43	0.95	1.19	3.58	0.72	0.96
Met	1.94	1.02	1.01	1.76	0.87	0.92	1.79	0.49	0.71	1.93	0.60	0.78	1.65	0.91	1.03	1.79	0.67	0.83
Lys	7.44	2.15	3.53	7.65	1.98	3.38	7.48	0.19	2.61	7.35	0.94	2.95	7.48	0.31	3.37	7.54	1.13	3.21
His	1.53	0.30	0.48	1.55	0.25	0.45	1.46	0.10	0.41	1.56	0.19	0.38	1.46	0.29	0.57	1.60	0.22	0.46
Phe	2.87	1.46	1.63	2.80	1.26	1.50	2.54	0.64	1.16	2.74	0.80	1.18	2.50	1.37	1.52	2.55	1.00	1.30
Trp	0.56	0.18	0.20	0.58	0.15	0.16	0.45	0.06	0.10	0.59	0.11	0.14	0.50	0.16	0.17	0.57	0.11	0.17
ΣEAAs	31.06	10.55	12.47	30.69	10.77	12.71	28.79	4.28	8.56	29.52	5.74	8.64	29.44	9.87	13.75	29.86	7.36	10.88
ΣAAs	81.03	17.13	21.87	79.94	17.55	21.37	78.65	9.34	16.24	76.97	10.52	15.85	78.00	16.54	21.68	80.32	12.31	18.73

Amino acid residues are denoted by their 3-letter code. Asx: Aspartic acid and asparagine, Glx: Glutamic acid and glutamine, EAAs: essential amino acids, nd: not detected, -: not determined.

**Table 3 marinedrugs-19-00383-t003:** Physicochemical properties of blue whiting (*Micromesistius poutassou*) soluble protein hydrolysates (BW-SPHs) pre- and post-simulated gastrointestinal digestion (GI).

Sample Code	Degree of Hydrolysis (DH%)	Molecular Mass Distribution (%)				
		>10 kDa	5–10 kDa	2–5 kDa	1–2 kDa	<1 kDa
BW-SPH-A	43.19 ± 2.16 ^a^	0.32 ± 0.08	1.47 ± 0.08	6.26 ± 0.04	14.11 ± 0.01	77.86 ± 0.16
BW-SPH-A_GI	57.48 ± 2.72 ^AB^*	0.00 ± 0.00	0.04 ± 0.06	1.22 ± 0.02	6.88 ± 0.23	91.87 ± 0.30
BW-SPH-B	45.78 ± 2.91 ^a^	0.99 ± 0.04	2.79 ± 0.03	9.10 ± 0.15	15.94 ± 0.12	71.19 ± 0.33
BW-SPH-B_GI	61.51 ± 2.66 ^AB^*	0.00 ± 0.00	0.13 ± 0.03	1.82 ± 0.02	8.36 ± 0.02	89.70 ± 0.07
BW-SPH-C	27.82 ± 1.11 ^c^	2.45 ± 0.05	5.56 ± 0.06	15.97 ± 0.07	20.48 ± 0.04	55.55 ± 0.13
BW-SPH-C_GI	55.37 ± 1.83 ^B^*	0.00 ± 0.00	0.27 ± 0.01	2.77 ± 0.02	11.38 ± 0.03	85.58 ± 0.01
BW-SPH-D	36.59 ± 0.87 ^b^	0.25 ± 0.05	1.49 ± 0.04	7.02 ± 0.03	16.67 ± 0.06	74.57 ± 0.08
BW-SPH-D_GI	56.13 ± 3.95 ^B^*	0.00 ± 0.00	0.09 ± 0.03	1.35 ± 0.06	7.99 ± 0.02	90.58 ± 0.11
BW-SPH-E	42.97 ± 3.30 ^a^	1.49 ± 0.02	3.31 ± 0.02	9.85 ± 0.05	15.62 ± 0.06	69.74 ± 0.13
BW-SPH-E_GI	65.23 ± 1.05 ^A^*	0.00 ± 0.00	0.19 ± 0.04	2.03 ± 0.05	8.54 ± 0.09	89.54 ± 0.15
BW-SPH-F	36.53 ± 2.05 ^b^	1.33 ± 0.02	3.71 ± 0.03	12.57 ± 0.06	19.45 ± 0.02	62.94 ± 0.11
BW-SPH-F_GI	57.90 ± 4.25 ^AB^*	0.00 ± 0.00	0.24 ± 0.04	2.40 ± 0.07	10.02 ± 0.07	87.34 ± 0.18

Mean ± SD (*n* = 3). Different lower case letters (^a,b,c^) indicate a significant difference with BW-SPH samples pre-simulated gastrointestinal digestion at *p* < 0.05. Different capital letters (^A,B^) indicate a significant difference with BW-SPH samples post- simulated gastrointestinal digestion at *p* < 0.05. * indicates a significant difference at *p* < 0.05 between pre- and post-simulated gastrointestinal digestion values.

**Table 4 marinedrugs-19-00383-t004:** Hydrolysis conditions used to generate blue whiting (*Micromesistius poutassou*) soluble protein hydrolysates A–F (BW-SPHs-A_F).

Sample Code	*Fish:Water*	Enzyme Preparation	E:S (% (*w*/*w*))	Temperature (°C)	Time (min)
BW-SPH-A	*2:1*	1 and 2	0.169 and 0.169	50	120
BW-SPH-B	*2:1*	2 and 3	0.169 and 0.169	50	120
BW-SPH-C	*1.7:1*	4 and 5	0.005 and 0.005	50	45
BW-SPH-D	*1.7:1*	4 and 5	0.900 and 0.290	50	60
BW-SPH-E	*2:1*	2 and 6	0.169 and 0.169	50	120
BW-SPH-F	*2:1*	7	0.340	50	120

(E:S) Enzyme to substrate ratio-weight of enzyme/weight of thawed minced fish.

**Table 5 marinedrugs-19-00383-t005:** Calculated amino acid score for the blue whiting (*Micromesistius poutassou*) soluble protein hydrolysates (BW-SPH-A_F) for different age categories.

Amino Acid	BW-SPH-A	BW-SPH-B	BW-SPH-C	BW-SPH-D	BW-SPH-E	BW-SPH-F
Amino acid ratio (infants (birth to 6 months))						
SAA	1.05	0.99	0.99	1.09	0.95	0.98
Trp	0.45 *	0.47 *	0.36 *	0.49 *	0.41 *	0.46 *
Thr	1.14	1.12	1.06	1.09	1.10	1.11
Val	0.96	0.96	0.85	0.90	0.95	0.89
Ile	0.83	0.84	0.69	0.76	0.85	0.76
Leu	0.82	0.81	0.79	0.82	0.79	0.80
AAA	0.88	0.76	0.67	0.79	0.71	0.71
Lys	1.47	1.53	1.49	1.51	1.53	1.50
His	0.99	1.02	0.96	1.06	0.98	1.04
Amino acid ratio (child (6 months to 3 years))						
SAA	1.28	1.22	1.21	1.33	1.16	1.20
Trp	0.90 *	0.94 *	0.73 *	0.99 *	0.83 *	0.92 *
Thr	1.62	1.59	1.51	1.55	1.56	1.58
Val	1.23	1.23	1.08	1.16	1.21	1.14
Ile	1.42	1.44	1.19	1.31	1.46	1.31
Leu	1.19	1.18	1.14	1.19	1.15	1.16
AAA	1.59	1.38	1.21	1.42	1.29	1.28
Lys	1.77	1.86	1.80	1.83	1.85	1.81
His	1.04	1.07	1.00	1.11	1.03	1.10
Amino acid ratio (older child, adolescent, adult)						
SAA	1.51	1.43	1.42	1.56	1.36	1.41
Trp	1.15 *	1.22 *	0.94 *	1.27 *	1.07 *	1.18 *
Thr	2.01	1.98	1.87	1.92	1.93	1.96
Val	1.32	1.32	1.16	1.24	1.30	1.22
Ile	1.52	1.54	1.27	1.40	1.56	1.39
Leu	1.29	1.28	1.24	1.28	1.25	1.26
AAA	2.02	1.75	1.54	1.81	1.63	1.62
Lys	2.11	2.21	2.14	2.18	2.19	2.15
His	1.30	1.34	1.25	1.39	1.28	1.37

* Amino acid score (1st limiting amino acid), Amino acid residues are denoted by their 3-letter code. SAA: sulphur amino acids, AAA: aromatic amino acids.

**Table 6 marinedrugs-19-00383-t006:** In vitro dipeptidyl peptidase-IV (DPP-IV) inhibitory and insulin secretory activity of blue whiting (*Micromesistius poutassou*) soluble protein hydrolysates (BW-SPHs) pre- and post-simulated gastrointestinal digestion (GI).

Sample Code	DPP-IV Inhibitory Activity IC_50_ (mg protein/mL)	Insulin Secretion (Fold Change Compared to Control)
BW-SPH-A	2.42 ± 0.04 ^b^	4.47 ± 0.27 ^d^
BW-SPH-A_GI	2.49 ± 0.07 ^AB^	4.80 ± 0.18 ^BC^
BW-SPH-B	2.37 ± 0.05 ^b^	6.00 ± 0.25 ^ab^
BW-SPH-B_GI	2.54 ± 0.05 ^AB^*	5.38 ± 0.19 ^A^
BW-SPH-C	2.90 ± 0.07 ^c^	4.66 ± 0.53 ^cd^
BW-SPH-C_GI	2.80 ± 0.04 ^C^	4.96 ± 0.21 ^AB^
BW-SPH-D	2.12 ± 0.03 ^a^	6.31 ± 0.21 ^ab^
BW-SPH-D_GI	2.39 ± 0.07 ^A^*	4.48 ± 0.14 ^C^*
BW-SPH-E	2.46 ± 0.03 ^b^	5.49 ± 0.20 ^bc^
BW-SPH-E_GI	2.63 ± 0.08 ^B^*	4.86 ± 0.10 ^BC^
BW-SPH-F	2.41 ± 0.03 ^b^	6.42 ± 0.30 ^a^
BW-SPH-F_GI	2.56 ± 0.07 ^AB^*	4.68 ± 0.13 ^BC^*

IC_50_: inhibitory concentration that inhibits recombinant human DPP-IV activity by 50%, mean ± SD (*n* = 3). Insulin secretion from clonal pancreatic BRIN-BD11 beta-cells is expressed as fold change compared to control (buffer/media containing 5.6 mM glucose alone), mean ± S.E.M. (*n* = 6). Different lower case letters (^a,b,c^) within a column indicates a significant difference between BW-SPH samples pre-simulated gastrointestinal digestion at *p* < 0.05. Different capital letters (^A,B^) within a column indicates a significant difference between BW-SPH samples post-simulated gastrointestinal digestion at *p* < 0.05. * indicates a significant difference at *p* < 0.05 between pre- and post-simulated gastrointestinal digestion values.

## Data Availability

Data are contained within the article or in Appendix A.

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
