# Peer review of "Physicochemical, Nutritional and In Vitro Antidiabetic Characterisation of Blue Whiting (Micromesistiuspoutassou) Protein Hydrolysates"

_marinedrugs, 2021, doi:10.3390/md19070383_

Round 1

Reviewer 1 Report

This study investigates nutritional, physicochemical, and in vitro anti-diabetic characteristics of six blue whiting solube protein hydrolysates (BW-SPHs). This is an interesting and well-designed work, which reports potentials of these BW-SPHs as a source of protein for human nutrition and health. However, it remains a few issues. My comments are as following:

  1. One of challenges for exploitation of fish as sources of functional ingredients at industrial scale is flavor and odor. Can the flavor and odor of these BW-SPHs be acceptable by users?
  2. Reference 41 is not relevant to in vitro measurement of insulinotropic effects of samples (Line 438).

3. Please indicate the origin of BRIN-BD11 cells and briefly describe the cell culture of these cells.

Author Response

Response: We thank the Reviewer for the careful consideration and review of our manuscript and the opportunity to clarify the points raised.

  1. One of challenges for exploitation of fish as sources of functional ingredients at industrial scale is flavor and odor. Can the flavor and odor of these BW-SPHs be acceptable by users?

 Response: This is a very pertinent point. While processes are being developed to try to improve the sensory properties of fish derived functional ingredients, the current situation is that fish protein hydrolysates are only accepted once they are provided in appropriate delivery vehicles/matrices. These matrices may also require the inclusion of specific masking agents. The following sentence has been included on Line 255 ‘However, due to their predominant marine taste and odour, the oral  ingestion of fish protein-derived hydrolysates  generally requires the utilisation of delivery matrices containing specific masking agents to render these ingredients organoleptically acceptable for consumers.’

  1. Reference 41 is not relevant to in vitro measurement of insulinotropic effects of samples (Line 438).

Response: Thank you for pointing out this error which was a mix up in the numbering of the references. Reference 41, which is relevant to the in vitro measurement of insulinotropic effects of samples should correspond to ‘McClenaghan, N.H.; Barnett, C.R.; Ah-Sing, E.; Abdel-Wahab, Y.; O'Harte, F.P.; Yoon, T.W.; Swanston-Flatt, S.; Flatt, P. Characterization of a novel glucose-responsive insulin-secreting cell line, BRIN-BD11, produced by electrofusion. Diabetes 1996, 45, 1132–1140.’ and reference 42 should correspond to ‘Flatt, P.R.; Bailey, C.J. Plasma glucose and insulin responses to glucagon and arginine in Aston ob/ob mice: Evidence for a selective defect in glucose-mediated insulin release. Horm. Met. Res. 1982, 14, 127–130.’ and not the other way around. This has now been rectified.

  1. Please indicate the origin of BRIN-BD11 cells and briefly describe the cell culture of these cells.

Response: As indicated in the previous response, reference [41] ‘McClenaghan, N.H.; Barnett, C.R.; Ah-Sing, E.; Abdel-Wahab, Y.; O'Harte, F.P.; Yoon, T.W.; Swanston-Flatt, S.; Flatt, P. Characterization of a novel glucose-responsive insulin-secreting cell line, BRIN-BD11, produced by electrofusion. Diabetes 1996, 45, 1132–1140.’ which contains details about the origin and culture conditions for BRIN-BD11 cells has been transferred to its correct place in the text.  Line 501 now read as ‘The insulinotropic effects of samples were measured in vitro using clonal pancreatic BRIN-BD11 cells which are a hybrid cell line formed following the electrofusion of immortal RINm5F cells with a primary culture of New England Deaconess Hospital rat pancreatic islet cells [41]. The cells were cultured at 37 ËšC in an atmosphere of 5% CO2 and 95% air in RPMI-1640 tissue culture medium containing 10% (v/v) fetal calf serum, antibiotics (100 U/mL penicillin, 0.1 mg/mL streptomycin) and 11.1 mM glucose as previously described [41].

Reviewer 2 Report

In the present report Harnedy-Rothwell et al demonstrated the analysis of protein hydrolysates derived from blue whiting and investigated their potential anti-diabetic actions. The results are novel and important in the field and the manuscript is well written. Nevertheless there is important information missing regarding the preparation of the different hydrolysates, their source and their differences that may explain the differences observed in the analyses.

Comments

  1. The authors mention the analysis of six BW-SPHs. The preparation process of each hydrolysate and how these differ in terms of source material and preparation process is not clear and should be clearly described.
  2. The authors should describe and explain the differences observed between the preparations based on the raw material and extraction process differences. Were there different enzymes used (and which?); was there a different process utilized (temperatures, time of incubation etc). These details are important for supporting the conclusions and findings and to provide scientific merit to the work. Without such detailed information the work is merely a descriptive presentation of results that cannot be reproduced by other scientists.
  3. Measuring the DPP4 inhibitory activity a positive control is missing such as a commercially available DPP4 inhibitor to allow the reader to compare the results with known inhibitors.
  4. The discussion is too long and does not focus on explaining the differences between the activity of the hydrolysates based on their preparation process and potential source. 
  5. As mentioned above, no information on the preparation procedures is provided in the material and methods section.

Author Response

Response: We thank the Reviewer for the careful consideration and review of our manuscript and the opportunity to clarify the points raised.

1. The authors mention the analysis of six BW-SPHs. The preparation process of each hydrolysate and how these differ in terms of source material and preparation process is not clear and should be clearly described.

Response: Thank you for this comment, we have now provided more details on the processes used to generate the six BW-SPHs. These have now been included by way of Table 6 ‘Hydrolysis conditions used to generate blue whiting (Micromesistius poutassou) soluble protein hydrolysates A-F (BW-SPHs-A_F)‘ along with a more detailed description in section 4.2. The food-grade microbial-derived proteolytic enzyme preparations used are proprietary to the company and we are unable to provide the specific details of same. Furthermore, the sentence on line 259 has been modified to indicate that the hydrolysates were generated from the same source material. ‘In this study six BW-SPHs generated from the same source material ….’

2. The authors should describe and explain the differences observed between the preparations based on the raw material and extraction process differences. Were there different enzymes used (and which?); was there a different process utilized (temperatures, time of incubation etc). These details are important for supporting the conclusions and findings and to provide scientific merit to the work. Without such detailed information the work is merely a descriptive presentation of results that cannot be reproduced by other scientists.

 Response: Thank you for your comment. As indicated in the previous response, additional details in relation to the method/process used to generate the six BW-SPHs have now been included. Furthermore, a number of sentences/paragraphs have now been included in the revised manuscript (see below) to highlight that each of the six hydrolysates generated are distinctly different and to demonstrate the potential link between the different characteristics observed and the hydrolysis conditions used in their generation.

Line 112: While BW-SPH-A, BW-SPH-B and BW-SPH-E were generated using similar hydrolysis conditions, except for the second enzyme in the two enzyme combination and had similar DH values their free amino acid profile varied (Table 2). In particular, BW-SPH-E had a reduced level of free lysine at 0.31 % (w/w) compared to BW-SPH-A and BW-SPH-B which had levels of 2.15 and 1.98 % (w/w), respectively. BW-SPH-A was shown to have lower levels of free isoleucine compared to the other two samples while BW-SPH-B was shown to have higher levels of free cysteine compared to BW-SPH-A and BW-SPH-E (Table 2). These differences in free amino acid profiles are most likely due to the specificity of the second enzyme preparation used for the generation of the three samples.

Line 147: As stated previously, BW-SPH-A, BW-SPH-B and BW-SPH-E were generated using similar hydrolysis conditions apart from the second enzyme in the two enzyme combination. This may explain why all three samples had similar DH values and molecular mass distribution profiles. Furthermore, the lower DH value and the level of components < 1kDa observed for BW-SPH-C may be explained by the lower E:S (0.1 %(w/w) minced fish) and hydrolysis time (45 min) used to generate this hydrolysate compared to that used for the generation of the other hydrolysates (E:S ≥ 0.338% (w/w) minced fish and an hydrolysis time of ≥ 1h).

Line 166: In particular, the intensity of the peaks between 0-8 min were significantly lower in BW-SPH-C than equivalent peaks in the other samples and specifically compared to those in BW-SPH-A, BW-SPH-B and BW-SPH-F (Supplementary Figure S1). In contrast the intensity of the peaks in the more hydrophobic region in particular at retention times >16 min are higher in BW-SPH-C compared to the other samples. A unique peak was also observed in BW-SPH-C at a retention time of 17.5 min. This may be associated with the lower extent of hydrolysis observed in BW-SPH-C. Furthermore, a unique peak was also seen at 15 min in the peptide profile of BW-SPH-F (Supplementary Figure S1).

Line 279: Furthermore, physicochemical data such as DH, molecular mass distribution, RP-UPLC and free amino profiles demonstrate that the variation in hydrolysis conditions resulted in the generation of BW-SPHs with distinctly different characteristics.

Line 540: The variation in hydrolysis conditions employed led to the generation of BW-SPHs with distinctly different characteristics.

3. Measuring the DPP4 inhibitory activity a positive control is missing such as a commercially available DPP4 inhibitor to allow the reader to compare the results with known inhibitors.

 Response: As indicated in the Methods section, the synthetic tripeptide Diprotin A was used as a positive control and it was seen to display an IC50 value of 5.00 µM. The IC50 value determined for Diprotin A can be used to compare the potency of peptides or other compounds when IC50 values are expressed in molar concentrations. However, the test hydrolysates under investigation contain hundreds if not thousands of peptides and therefore IC50 values are expressed in mg/ml. We do however use the positive control in hydrolysate assessment studies to compare our results to data reported for other hydrolysates which may have been assessed under different conditions. As shown from Line 337 it was used to explain why the DPP-IV inhibitory activity observed with the BW-SPHs herein were lower than that previously observed with BWSPH. ‘However, this difference may be due to the origin of the DPP-IV enzyme utilised in both studies. The present study used human-derived DPP-IV whereas the previous study used porcine-derived DPP-IV. It has been documented that human-derived DPP-IV is less active than its porcine-derived equivalent [21,22]. For example, the IC50 values determined previously for the reference DPP-IV inhibitory tripeptide, Ile-Pro-Ile (Diprotin A), was 3.49 ± 0.19 and 5.00 ± 0.03 µM for porcine and human-derived DPP-IV, respectively [21, 23].’

4. The discussion is too long and does not focus on explaining the differences between the activity of the hydrolysates based on their preparation process and potential source. 

Response: We acknowledge this comment. However, manuscript contains a significant quantity of data and each set of data deserves to be discussed adequately. 

5. As mentioned above, no information on the preparation procedures is provided in the material and methods section.

Response: A response to this comment has been given in responses 1 and 2 above

Reviewer 3 Report

This manuscript descibes the characterization of the physicochemical, nutritional and in vitro antidiabetic activity of blue whiting (Micromesistius poutassou) protein hydrolysates. Despite, it seems to be well-designed, there are two minor points:

  • regarding the HPLC analysis, it is not clear which compounds are evaluated/detected.
  • relatively to the articles cited about the methods used seems to be very old manuscripts (maybe could be added also more recent references regarding these methods used). 

Author Response

Response: We thank the Reviewer for the careful consideration and review of our manuscript and the opportunity to clarify the points raised.

1. Regarding the HPLC analysis, it is not clear which compounds are evaluated/detected.

Response: For RP-UPLC and GPC-HPLC analysis, detection was performed at 214 nm, which detects absorbance of the peptide bond. However, to ensure that the reader is clear on what is being evaluated/detected with these chromatographic techniques the sentence starting on Line 427 has been modified and now reads as followings: ‘Molecular mass distribution and hydrophobicity/hydrophilicity analysis were carried out by gel permeation-high performance liquid chromatography (GP-HPLC) and RP-UPLC, respectively, in order to determine the peptide profiles of the test samples [39, 40].’

2. Relatively to the articles cited about the methods used seems to be very old manuscripts (maybe could be added also more recent references regarding these methods used).

Response: The methodology employed in this study arises from peer-reviewed publications and we specifically referred to the source references for same throughout.

Round 2

Reviewer 1 Report

No comments